# Spatiotemporal Graph Indicators for Air Traffic Complexity Analysis

**Ralvi Isufaj [1,\*]**, **Thimjo Koca [2]** and **Miquel Angel Piera [1]**

1 Logistic and Aeronautics Group, Department of Telecommunications and System Engineering, Autonomous University of Barcelona, 08202 Sabadell, Spain; miquelangel.piera@uab.cat
2 Intelligent & Autonomous Systems Group, CWI, 1098 Amsterdam, The Netherlands; thimjo.koca@cwi.nl
\* Correspondence: ralvi.isufaj@uab.cat

**Abstract:** There has been extensive research in formalising air traffic complexity, but existing works focus mainly on a metric to tie down the peak air traffic controllers workload rather than a dynamic approach to complexity that could guide both strategical, pre-tactical and tactical actions for a smooth flow of aircraft. In this paper, aircraft interdependencies are formalized using graph theory and four complexity indicators are described, which combine spatiotemporal topological information with the severity of the interdependencies. These indicators can be used to predict the dynamic evolution of complexity, by not giving one single score, but measuring complexity in a time window. Results show that these indicators can capture complex spatiotemporal areas in a sector and give a detailed and nuanced view of sector complexity.

**Keywords:** air traffic complexity; spatiotemporal indicators; complexity metrics; graph theory





## 1. Introduction

The mission of air traffic management (ATM) is to make air traffic possible by means of efficient, environmentally friendly and socially valuable systems [1,2]. At the heart of the current ATM system at the tactical level are human air traffic controllers (ATCo) who control airspace units known as sectors. The biggest responsibility of ATCos is guaranteeing safety, which means they must issue instructions to pilots, monitor traffic to maintain safety distances and so on. The ability of controllers to effectively fulfil these duties is constrained by their workload, which can be defined as the mental and physical work done by controllers to manage traffic [3]. Therefore, it is important to keep ATCo workload at acceptable levels.

Controller workload is not easy to predict or estimate and is related to various factors which can be qualitative and quantitative [4,5]. However, several studies [5–7] state that airspace complexity accounts for a large portion of workload. Complexity has been a prominent topic of research in ATM (see Section 2 for a more detailed review). A predictor of complexity is key not only to support ATCos, but also for a more sustainable and efficient air traffic management system in which complexity effects could be mitigated at early stages. However, there are several drawbacks to already existing metrics. First of all, the majority of metrics do not give a detailed view of complexity. For instance, the most widely used metric is Aircraft Density [8,9], which measures the number of aircraft flying in a sector. This metric, however, is shown not to be adequate in capturing ATCo workload. For instance, Delahaye and Puechmorel [4] take the example of sector capacity, which is defined as the maximum number of aircraft that can be accommodated in a given time period [10]. They observe cases where controllers accept more than the actual capacity and cases where they refuse traffic even if capacity has not been reached.

Other methods attempt to measure ATCo workload directly [11–13]. These methods, however, tend to be intrusive to the workflow of controllers, in addition to having huge

computational costs, which make them not suitable for practical uses. The majority of these metrics do not consider how complexity evolves in time.

Another area where complexity metrics can impact ATCo workload and as a consequence ATM capacity, are conflict detection and resolution (CD & R) [14–17] decision support tools. A desirable characteristic of such solvers will be the quality of solutions in terms of complexity, which means that a conflict resolution that leaves the sector in a more complex state should be discouraged in favour of solutions that ideally lead to lower complexity, while preserving safety.

In this work, air traffic is modelled through graph theory and complexity is defined as the connectivity of the graph. Four indicators are proposed that represent different aspects aircraft interaction. The indicators measure several structural properties of the (traffic as a) graph, thus giving different insights into complexity. *Edge density* measures the size of the graph with respect to a fully connected graph with maximal weights, which identifies the aircraft that will create interdependencies. *Strength* measures the severity of the interdependencies, as aircraft that are closer present a more complex situation. The *Clustering Coefficient* provides information regarding the neighbourhood of each aircraft and the *Nearest Neighbour Degree* identifies if interdependent aircraft point to neighbours that may also have various interdependencies. As results show, each of the indicators is necessary and the information gathered from them can capture different shapes of interdependencies. This is an important step towards tackling hotspots, which are complex spatiotemporal areas in a sector, rather than simply solving conflicts. Furthermore, the evolution of complexity through time is considered, which gives a detailed view of how aircraft interactions change in time.

The rest of this paper is organizes as follows: in Section 2, we elaborate on existing complexity metrics and discuss some of their drawbacks. The theoretical background and modelling of air traffic using graph theory is presented in Section 3. In Section 4, we present the proposed complexity indicators. We evaluate the indicators in illustrative examples in Section 5, while results obtained from real traffic are presented and discussed in Section 6. In Section 7, we draw conclusions and propose steps for further research.

## 2. Related Work

First of all, there have been previous works that model air traffic as a graph. Koca et al. [18] effectively transform traffic to a graph representation by identifying relevant aircraft to a conflict by means of spatiotemporal regions. However, their work is applicable only to conflict resolution. Furthermore, their graph analysis requires the presence of pairwise conflicts and is not intended to provide a dynamic complexity metric that considers traffic evolution.

There are many different ATM complexity metrics used in the literature. As is the case with this paper, the bulk of them focus on measuring sector complexity at the tactical level.

The complexity of a certain unit of airspace has been usually linked to controller workload [6]. This seems quite intuitive, the more work a controller has to do, the more complex the situation is. However, measuring it is not trivial. Workload is highly subjective and there is no definite consensus as to what constitutes it [11]. Research on complexity metrics can be generally put into two groups: metrics that correlate workload with certain physical attributes of the airspace and metrics that attempt to measure directly the workload on the controller.

The most common metric from the first group is Aircraft Density [8,9], defined as the number of aircraft at the time of measurement. Another common metric is Dynamic Density (DD) [19–21]. There are several ways DD has been defined in the literature, but the underlying core idea is to define complexity as the weighted sum of several attributes. Examples of such attributes include the number of aircraft, number of cruising, ascending and descending aircraft, speed, heading change, number and time to conflicts, etc. Interval Complexity [22], is similar to DD but is calculated as the average over a 5–10 min window. However, they do not give an evolution of complexity in time.

Another metric that measures complexity through geometrical features is Fractal Dimension [23]. Complexity is calculated through the degrees of freedom of an aircraft given its route and constraints. Nonetheless, it comes with strong dependencies on airspace structure, requiring it to be made up of piece-wise linear segments.

Delahaye et al. [4] propose two distinct ways of measuring air traffic complexity. Firstly, they consider geometrical properties in order to build a new complexity coordinate system in which sector complexity through time is presented. Secondly, they represent air traffic as a dynamic system in order to yield an intrinsic measure of complexity through Kolmogorov entropy.

Wang et al. [24] propose describing air traffic situations using the theory of complex networks. They provide a complexity vector, which is comprised from several indicators. Situations are then classified as 'low-complexity', 'medium-complexity' and 'high-complexity' depending on vector values.

Their approach bears some similarities to this paper; however, as we will explain, there are some key differences. First of all, they consider two aircraft interdependent only if there is a potential conflict between them, while conflicts affect the complexity of a situation, they are not the only source of it [4,7,8,18]. Furthermore, they do not consider the severity of interdependencies. In this work, two aircraft are interdependent if they are close enough horizontally and vertically. Additionally, we consider the severity of interdependencies with a conflict having maximal severity (see Section 3 for a detailed description).

The second group of complexity metrics consists of approaches that measure the cognitive workload controllers experience. They include methods such as measuring ocular activity [11], brain imaging [12] and electroencephalography [13].

COTTON [25] is a project funded by SESAR JU that investigates many aspects that affect complexity (geometric and cognitive), which they call complexity generators. Different airspace configurations and time horizons are considered. They identify the most influencing generators and combine them. Furthermore, they consider uncertainty which they model through Bayesian networks.

Both groups of approaches have several drawbacks. Most of the metrics in the first group take a very simple point of view (e.g., aircraft density), while this means that those metrics can be computed quickly, it also means that the information they provide is not comprehensive. These metrics are not sufficiently expressive, as providing only a complexity score does not give the controllers a detailed reasoning why a situation is complex or not. The geometrical approach by Delahaye et al. overcomes these drawbacks; however, it is not clear how to interpret the proposed coordinate system. Furthermore, their second approach based on dynamic system theory requires a calibration time and suffers from higher computational costs. Moreover, it is more suited for measuring flow complexity.

On the other hand, the second group of approaches tends to use methods that are quite intrusive to the workflow of the controller. Additionally, they measure workload on each controller individually, which will require an unknown calibration time to specific controllers. Finally, many of these approaches show complexity only for a fixed point in time. This is another drawback of such approaches, as the evolution of traffic through time is an important indicator of how complex a situation is.

Automation in ATM requires a better understanding of potential threats that can impact traffic and as such, complexity metrics should provide elaborated insights about the right mitigation measures to apply.

## 3. Air Traffic Modelled as a Graph

In this section, a formal definition of graphs is given and some basic attributes important to this paper are introduced [26].

### 3.1. Some Definitions of Graph Theory

**Definition 1.** *An undirected graph $G = (V, E)$ is a mathematical structure that consists of a set $V$ of elements called vertices and a set $E$ of pairs of vertices called edges.*

Let $e = (a, b)$ be an edge of $G$. Then $e$ joins the two vertices $a, b \in V$ and is called *incident* of $a$ and $b$. In turn, those vertices are called the *endpoints* of $e$ and they are *adjacent* to each other.

The degree of a vertex of a graph is the number of edges that are incident to the vertex. The *order* of a graph $G = (V, E)$ is $|V|$, while the *size* of the graph is $|E|$.

A triplet is a group of three vertices that are fully connected, i.e., any pair of the three vertices are connected by an edge. We denote with $\mathcal{T}$ the set of all triplets in the graph.

$H = (U, F)$ is a *subgraph* of $G$ if the vertices and edges of $H$ are subsets of the vertices and edges of $G$, i.e., $U \subseteq V$ and $F \subseteq E$.

Similar to undirected graphs, directed graphs (digraphs) can also be defined. The difference in definition, is that in the case of directed graphs, $E$ is now a set of ordered edges. All attributes can be adapted.

Graphs can be weighted or unweighted. In the case of weighted graphs, each edge is assigned a number (the weight), for example between 0 and 1. If no edge connects two vertices, the weight is 0.

An important attribute of weighted graphs is the strength of a vertex. Its definition is analogous to the degree, but it takes into consideration the weights:

$$s(i) = \sum_{j=1}^{N} w_{i,j} \tag{1}$$

In addition to the visual representation of a graph, there are several ways a graph can be described. The most common way is the adjacency matrix $A$, which for unweighted graphs is:

$$A = \begin{pmatrix} a_{1,1} & a_{1,2} & \cdots & a_{1,n} \\ \vdots & \vdots & \vdots & \vdots \\ a_{n,1} & a_{n,2} & \cdots & a_{n,n} \end{pmatrix} a_{i,j} = \begin{cases} 1, \text{ if } (v_i, v_j) \in E \\ 0, \text{ otherwise} \end{cases} \tag{2}$$

The idea behind this representation is this: build a matrix with all possible edges between all vertex pairs of a graph. If an edge is actually present in the graph, then the entry in the matrix is 1, otherwise it is 0. Similarly, a weighted graph can also be represented with an adjacency matrix:

$$A_w = \begin{pmatrix} a_{1,1} & a_{1,2} & \cdots & a_{1,n} \\ \vdots & \vdots & \vdots & \vdots \\ a_{n,1} & a_{n,2} & \cdots & a_{n,n} \end{pmatrix} a_{i,j} = \begin{cases} w_{i,j}, \text{ if } (v_i, v_j) \in E \\ 0, \text{ otherwise} \end{cases} \tag{3}$$

In the weighted case, the entries for the edges present in the graph correspond to the weight $w_{i,j}$ of the edge.

In the case of an undirected graph, the adjacency matrix is symmetric, while for directed graphs this does not hold.

### 3.2. Modelling Aircraft Spatiotemporal Interdependencies Using Graphs

In this section, we describe the model that formalizes spatiotemporal interdependencies of en-route traffic as a weighted undirected graph.

A correct definition of a graph requires having a set of vertices and a set of edges. In our case, vertices are the set of aircraft present in a sector at a certain time step. Therefore, we extend graph attribute to the time domain, by defining each of them per time step. The set of edges will be the interdependencies between each pair of aircraft for the time step. We define these interdependencies based on the distance between two aircraft. More concisely, if two aircraft are closer than a certain threshold, then there will be an edge between these two aircraft. The closer these aircraft are, the bigger the effect they have on each other will be, i.e., the stronger the weight of the edge connecting the pair of aircraft. If the two aircraft are in a conflict, which means they are closer than the standard safety

distance (5 NM horizontally and 1000 feet vertically) the effect they have on each other is maximal.

In this work, the weights are set following this rationale. We calculate horizontal and vertical distance (weight) between all pairs of aircraft. An interdependency will be added only when two aircraft are close enough horizontally and vertically. Weights are normalized to be between 0 and 1 and the final weight is the average of the horizontal and vertical interdependency. Formally, this is:

$$wh_{i,j}(t) = \begin{cases} 1 \text{ if } dh_{i,j}(t) \leq H \\ 0 \text{ if } dh_{i,j}(t) \geq thresh_h \\ \frac{thresh_h - dh_{i,j}(t)}{thresh_h - min_h} \text{ otherwise} \end{cases} \tag{4}$$

$$wv_{i,j}(t) = \begin{cases} 1 \text{ if } dv_{i,j}(t) \leq V \\ 0 \text{ if } dv_{i,j}(t) \geq thresh_v \\ \frac{thresh_v - dv_{i,j}(t)}{thresh_v - min_v} \text{ otherwise} \end{cases} \tag{5}$$

$$w_{i,j}(t) = \begin{cases} \frac{wh_{i,j}(t) + wv_{i,j}(t)}{2} \text{ if } wh_{i,j}(t) > 0 \text{ \& } wv_{i,j}(t) > 0 \\ 0 \text{ otherwise} \end{cases} \tag{6}$$

where $wh_{i,j}(t)$ and $wv_{i,j}(t)$ are the horizontal and vertical weights at time $t$, $dh_{i,j}(t)$ and $dv_{i,j}(t)$ are the horizontal and vertical distance of two aircraft at time $t$, $H$ and $V$ are the horizontal and vertical safety distances and $thresh_h$ and $thresh_v$ are the horizontal and vertical thresholds. Such a definition of the interdependencies implies that they are undirected, which means that also the graph they define is undirected. Furthermore, the interdependencies are defined for a time step, therefore through their evolution in time, we are able to capture directional information such as heading. For instance, if the two aircraft that have an interdependency between them are moving towards each other, the weight of the interdependency would increase.

## 4. Complexity Indicators

There is a wealth of research present on graph complexity [27–34]. Many different definitions of complexity exist, depending on what aspects of graphs are being studied.

In this work, graph complexity and in turn sector complexity, is defined as the connectivity of the graph. Furthermore, by modelling traffic as a weighted graph, we inherently take into consideration the severity of interdependencies.

There are several ways the connectivity of a graph can be measured. Research that applies graph theory to practical problems [33,34] shows that connectivity indicators that combine topological information with the weight distribution of the graph are able to provide broad and detailed information. In this work, four indicators are formally defined and illustrated: *edge density*, *strength*, *clustering coefficient* and *nearest neighbour degree*.

### 4.1. Edge Density

Edge density (ED) measures how many edges the graph has, compared to the number of edges in a fully connected graph of the same size. As we are dealing with weighted graphs, the weights are considered. Formally, ED is given as follows:

$$ED(G, t) = \frac{\sum_{(i,j) \in E} w_{i,j}(t)}{A(V_t)}, A(V_t) = \frac{|V_t|(|V_t| - 1)}{2} \tag{7}$$

$|V_t|$ denotes the number of vertices in the graph (i.e., the number of aircraft present in the sector) at time step $t$ and $A(V_t)$ is the number of all possible edges. From the definition, it follows that this indicator can take values from 0 to 1.

ED refers to the whole graph, and not specific vertices, making it a global connectivity measure. It relies on the concept that traffic geometries tend to be complex when there are more interdependencies between aircraft.

Figure 1 shows an arbitrary sector. The only interdependency exists between $AC_3$ and $AC_4$. However, as there are four aircraft in the sector, the potential number of interdependencies is six, therefore the ED score for this sector is $\frac{1}{6}$.

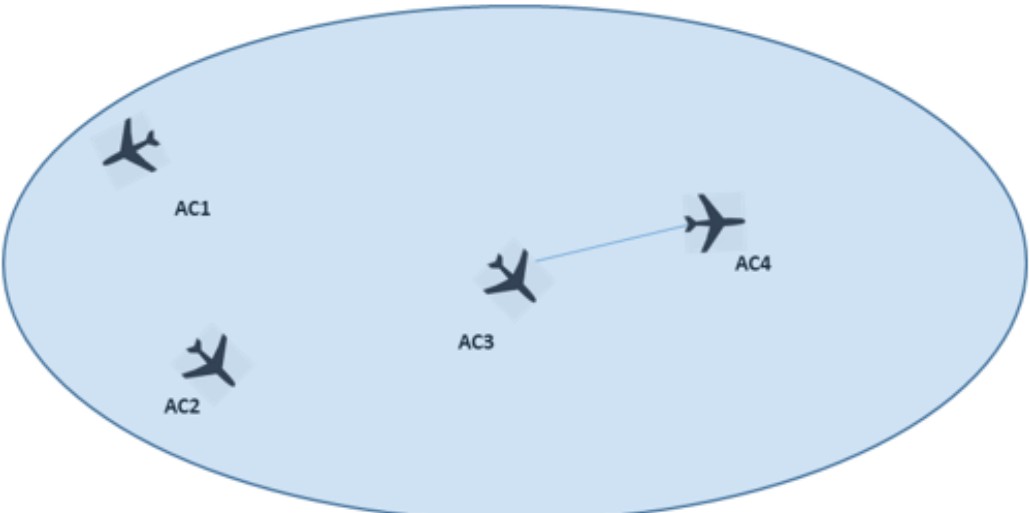

**Figure 1.** The edge density indicator.

*4.2. Strength*

In graph theory, the definition of strength is obtained by extending the definition of vertex degree to account for the weights of the edges. This indicator gives each aircraft its own score, and a global score is measured by taking the average of all aircraft in the graph. Formally, it is given as follows:

$$s(i,t) = \sum_{j=1}^{N} w_{i,j}(t) \tag{8}$$

Strength is a natural measure of the importance or centrality of a vertex in the graph. This indicator measures the strength of the vertices in terms of the total weight of their connections. In the proposed model, it quantifies how tight interdependencies of each aircraft are. The "stronger" an aircraft is, the more interdependent it is with other aircraft, the more complex it can be considered. This indicator is shown in Figure 2 where an arbitrary sector is shown in different time steps, There are three aircraft in the sector. $AC_1$ and $AC_2$ are moving closer while $AC_1$ and $AC_3$ are moving away from each other. As a result, the strength of AC3 decreases through time; however, as $AC_1$ and $AC_2$ are getting closer, their strength score increases. With $AC_1$ having an increasingly stronger interdependency with $AC_2$, while its interdependency with $AC_3$ grows slightly weaker, strength increases in time for the whole sector. Strength can take values from 0 to $|V_t| - 1$, which happens in the case of a fully connected graphs with maximal weights.

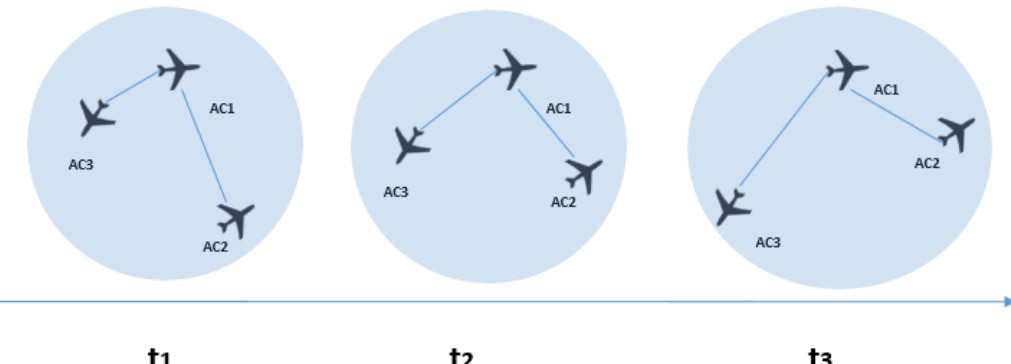

**Figure 2.** The strength indicator.

### 4.3. Clustering Coefficient

The clustering coefficient (CC) measures the local cohesiveness. This indicator provides information regarding the neighbourhood of each vertex. It takes into account the weight of the clustered structure found in triplets. For each vertex $i$, CC counts the number and the weight of triplets (see: Section 3) formed in the neighbourhood of $i$. Formally, the clustering coefficient of a vertex $i$, is calculated as follows:

$$CC(i,t) = \frac{\sum_{j,k} (w_{i,j}(t) + w_{j,k}(t))}{2 \cdot (s(i,t)(deg(i,t) - 1)}, \forall (i,j,k) \in \mathcal{T}(t) \tag{9}$$

where $s(i,t)$ is the strength of the current vertex, $deg(i,t)$ is the degree of the vertex at time step $t$ and $\mathcal{T}(t)$ is the set of triplets present at time $t$. CC scores range from 0 to 1.

If aircraft that are very tight with each other form clusters, then the situation will be more complex than if the clusters were formed by aircraft that form edges with smaller weights.

The clustering coefficient indicator is illustrated in Figure 3. In $t_1$, the three aircraft form a cluster. Such a configuration of aircraft represents a situation where they are tightly connected. It can be observed that $AC_3$ is moving away from the two other aircraft, thus breaking the cluster. In the first timestep, the controller would need to be concerned with all of the aircraft present in the sector, while in the second timestep, an interdependency exists only between $AC_1$ and $AC_2$. This illustration shows how only measuring the strength is not enough to give a rich picture of complexity, but different topological characteristics need to be examined.

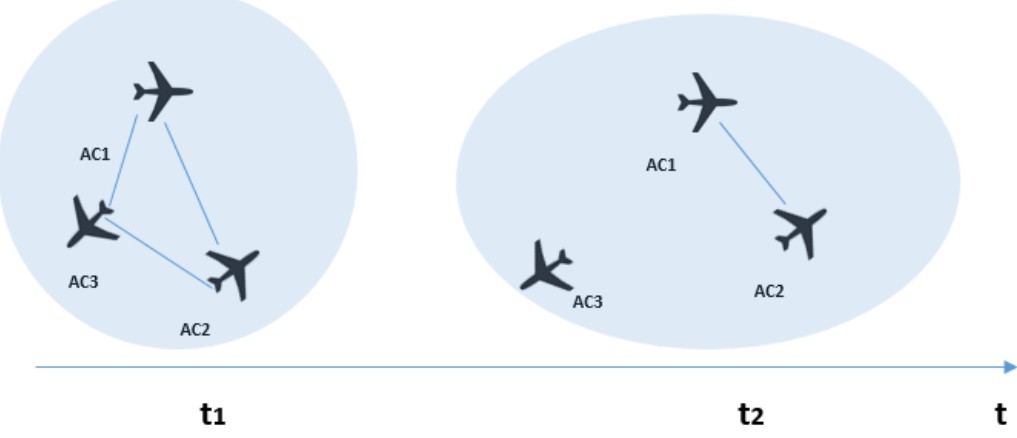

**Figure 3.** The CC indicator.

### 4.4. Nearest Neighbour Degree

Nearest Neighbour Degree (NND) calculates a local weighted average of the nearest neighbour degree of each aircraft according to the edge weights. Formally, it is defined as:

$$NND(i,t) = \frac{\sum_{j=1}^{N} w_{i,j}(t) deg(j,t)}{s(i,t)} \tag{10}$$

Such a definition implies that when edges with larger degrees are pointing to neighbors with higher degrees, the situation is more complex. Similar to Strength and CC, NND is also a local measure, and the global measure is calculated by averaging over all vertices. The NND scores range from 0 to $|V_t| - 1$. In the case of sector complexity, the more tightly connected a neighbour of an aircraft is to other aircraft, the more likely it is for a situation to arise that requires closer monitoring or potential ATCOs interventions. This indicator is illustrated in Figure 4.

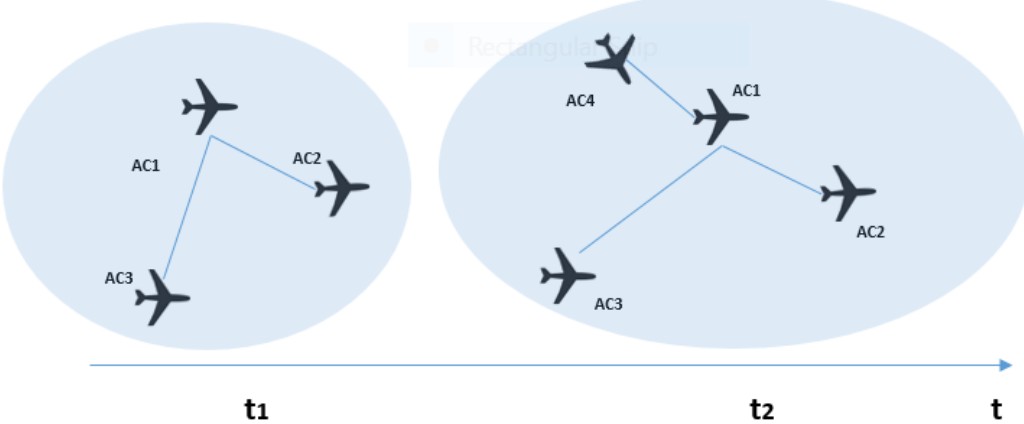

**Figure 4.** The NND indicator.

In $t_2$, $AC_4$ joins the sector, and immediately creates an interdependency with $AC_1$. The strength of $AC_4$, is therefore equal to its interdependency with $AC_1$. This would indicate that its complexity is quite low; however, $AC_1$ forms interdependencies with the rest of aircraft in the sector. As such, any potential manoeuvrers applied to $AC_4$ could affect $AC_1$, which in turn would affect the rest of the aircraft. Such a situation illustrates how interdependencies between aircraft where at least one has a high degree can lead to a highly complex situation. In this case, no clusters are formed, therefore the CC indicator does not change its value. This shows how clustering is not the only topological characteristic that captures the complexity of a situation, but all indicators are required to give a nuanced view of complexity.

Algorithm 1 shows the pseudocode of the procedure to calculate the complexity of a given airspace. The inputs are the airspace (e.g., coordinates of sector boundary), the time window for which to calculate the complexity, sampling time (e.g., measure complexity every 15 s) and the thresholds to determine interdependencies. The scores of the indicators are initialized as empty lists. Then, for each sampling time, until the duration of the time window has been reached, a snapshot of the traffic in the airspace is taken. After that, the graph is generated taking into account the threshold values for interdependencies. Then the indicator values for the current time are calculated using Equations (7)–(10) and the values for each indicator are returned.

---

**Algorithm 1** Calculate Complexity Algorithm

---

$\quad$**procedure** COMPLEXITY($airspace, t_{window}, t_{sample}, threshold$)
$\quad\quad$ $ED \leftarrow \varnothing$
$\quad\quad$ $CC \leftarrow \varnothing$
$\quad\quad$ $NND \leftarrow \varnothing$
$\quad\quad$ $Strength \leftarrow \varnothing$
$\quad\quad$ $t_{curr} \leftarrow 0$
$\quad\quad$ **while** $t_{curr} <= t_{window}$ **do**
$\quad\quad\quad$ $traffic_{t_{curr}} \leftarrow$ GET-TRAFFIC($t_{curr}, airspace$)
$\quad\quad\quad$ $graph \leftarrow GENERATE - GRAPH(traffic_{t_{curr}}, threshold)$
$\quad\quad\quad$ $ED_{t_{curr}} \leftarrow CALC - ED(graph)$
$\quad\quad\quad$ $CC_{t_{curr}} \leftarrow CALC - CC(graph)$
$\quad\quad\quad$ $NND_{t_{curr}} \leftarrow CALC - NND(graph)$
$\quad\quad\quad$ $Strength_{t_{curr}} \leftarrow CALC - Strength(graph)$
$\quad\quad\quad$ $ED \leftarrow ED \cup ED_{t_{curr}}$
$\quad\quad\quad$ $CC \leftarrow CC \cup CC_{t_{curr}}$
$\quad\quad\quad$ $NND \leftarrow NND \cup NND_{t_{curr}}$
$\quad\quad\quad$ $Strength \leftarrow Strength \cup Strength_{t_{curr}}$
$\quad\quad\quad$ $t_{curr} \leftarrow t_{curr} + t_{sample}$
$\quad\quad$ **end while**
$\quad\quad$ **return** $ED, CC, NND, Strength$
$\quad$**end procedure**

---

## 5. Test Scenarios

In this section, we illustrate how the indicators behave by means of synthetic trajectories. After that, Miles-in-Trail scenarios and real traffic are used to verify the expected benefits.

### 5.1. Illustrative Examples

To provide a better understanding of the complexity indicators, three synthetic scenarios are shown.

Figure 5 shows a graph with relatively high connectivity, while Figure 6 shows a less connected graph. Both graphs have the same number of vertices and common edges have the same weight. $G_1$ has more edges.

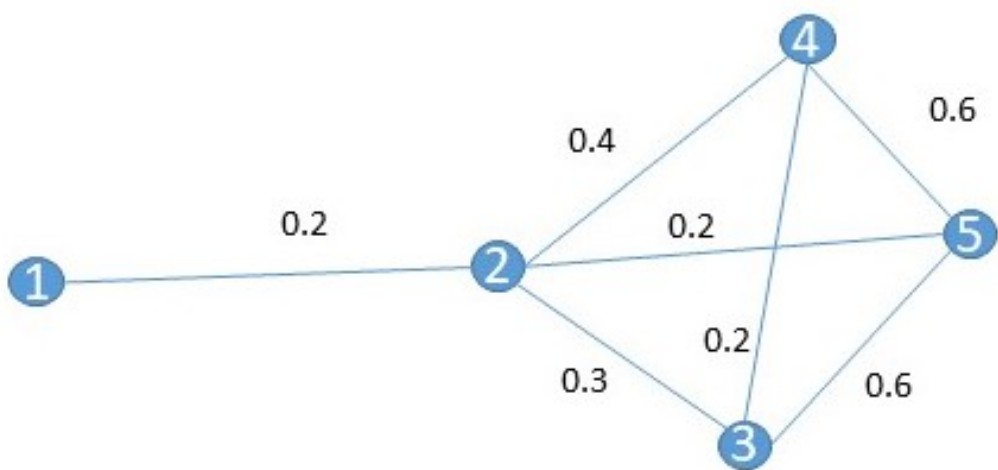

**Figure 5.** A complex graph $G_1$.

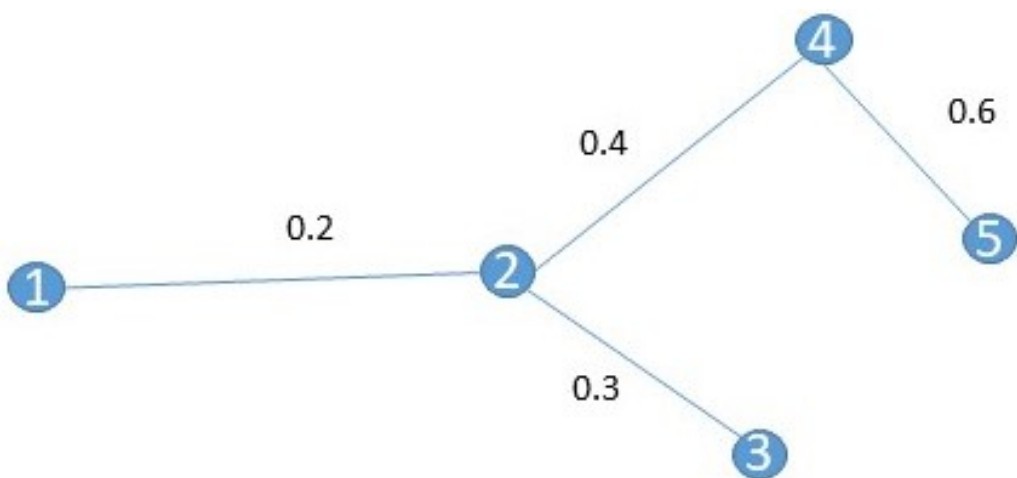

**Figure 6.** A simpler graph $G_2$.

Table 1 shows the indicator scores for $G_1$, while Table 2 shows them for $G_2$. As expected, because $G_1$ has more edges, it has a higher ED score.

**Table 1.** Complexity indicators for $G_1$.

|       | ED   | Strength | CC   | NND  |
|-------|------|----------|------|------|
| AC1   |      | 0.2      | 0    | 4    |
| AC2   |      | 1.1      | 0.2  | 2.63 |
| AC3   |      | 1.1      | 0.33 | 3.27 |
| AC4   |      | 1.2      | 0.33 | 3.33 |
| AC5   |      | 1.4      | 0.33 | 3.14 |
| Average | 0.25 | 1.0    | 0.24 | 3.28 |

**Table 2.** Complexity indicators for $G_2$.

|       | ED   | Strength | CC   | NND  |
|-------|------|----------|------|------|
| AC1   |      | 0.2      | 0    | 3    |
| AC2   |      | 0.9      | 0    | 1.44 |
| AC3   |      | 0.3      | 0    | 3    |
| AC4   |      | 1.0      | 0    | 1.8  |
| AC5   |      | 0.6      | 0    | 2    |
| Average | 0.15 | 0.6    | 0    | 2.25 |

In the case of CC, there are no clusters formed for $G_2$ Therefore, CC for each of its vertices is 0. This is not the case for $G_1$, where there are several triplets, i.e., clusters, thus the average CC is not zero, specifically 0.24.

As it has been previously mentioned, NND measures how connected the neighbours of a vertex are. Let us consider $AC_2$. By removing an edge between $AC_3$ and $AC_5$ (which has a relatively big weight), we can see the difference in NND score. The NND impacts the measures taken to lower complexity. If $AC_2$ and $AC_3$ are moving closer to each other, the strength of their interdependency will increase. Moreover, given that there is an interdependency between $AC_3$ and $AC_5$, the situation will be more complex. Having this information, a change in trajectory would be proposed in order to decrease complexity.

Figure 7 shows a more realistic example scenario. There, four aircraft are moving towards each other. In this example, they are all A320, flying at 30,000 feet, with a speed of 300 kts. The distance between the furthest aircraft is around 240 NM, while the distance between the closer aircraft is around 120 NM. The threshold for interdependencies is set to 100 NM. The simulation is run for 20 min. The figure shows snapshots of the traffic in

different time stamps, at the beginning, at 8 min and at 16 min. At each point in time, the graph and the indicator scores are shown, while evaluating the merits of the proposed indicators is paramount, an important aspect to investigate is the way the information is presented to the controllers. In this instance, we present a simple and intuitive way which presents the most crucial aspects of the graph. However, more research is needed to determine the best way such information must be presented.

Figure 8 shows the evolution of indicator in the simulation time. For the first 3 min, the scores are all zero, as the aircraft have not yet started creating interdependencies amongst them. The first interdependencies are created between $AC_1 - AC_3$ and $AC_2 - AC_4$. In this example, there was no intervention to separate the aircraft, therefore the peak for ED and Strength is reached around the 15 min mark, where all aircraft are in conflict with each other, as demonstrated by the ED value of 1, which is the theoretical maximum. Interestingly, the peak values of CC and NND are reached around the 10 min mark. Such a result indicates that all aircraft now have interdependencies with each other, shown by the value of NND being 3. This example illustrates how complex scenarios can occur even before the start of conflicts, as evidenced by the peak of CC and NND happening before the peak of ED and Strength.

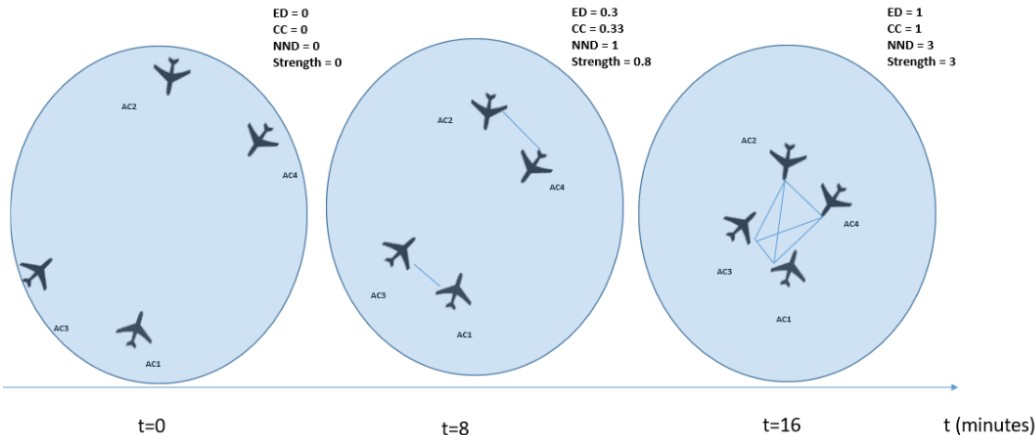

**Figure 7.** A more realistic example scenario.

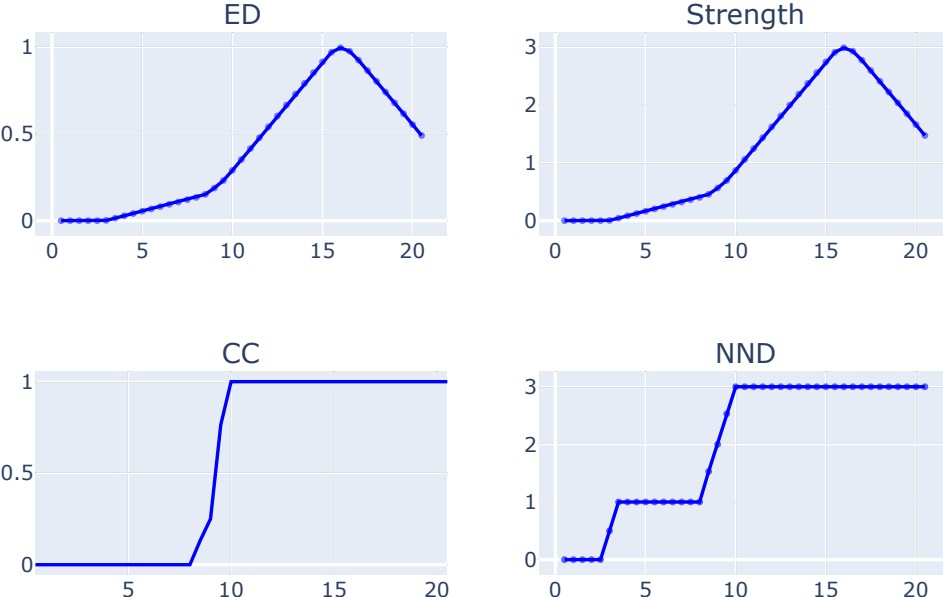

**Figure 8.** Indicator scores for the example scenario.

### 5.2. Miles-in-Trail

A realistic example is the intersection of two Miles-in-Trail (MiT) flows. This way of traffic organisation is well accepted among controllers to reduce air traffic complexity. In MiT, traffic is structured into flows of aircraft following the same path. Flights in the same path are separated by a certain distance and their speed is regulated. These flows create queues of aircraft that are easier to manage by controllers [15]. In this section, we show different conflict scenarios and how they would be solved by different CR algorithms and we will illustrate how the values of each complexity indicator evolve during that time.

To do so, we adapt the scenario of Breil et al. [15]. They consider a flow that goes through waypoints LMG, MEN and MRM and another that goes through waypoints TOU, MEN and LYS. Aircraft are generated every 100 s from LMG and TOU at 20,000 ft. The distance between aircraft in the same flow is kept at 14.14 NM. The interdependency threshold is set at 15 NM, so aircraft at the same flow always form interdependencies amongst them. Using these parameters, conflicts are induced in crossing aircraft from different flows.

In their work, Breil et al. [15] model air traffic as a multi-agent system and propose an algorithm to solve conflicts by local speed changes (henceforth referred to as MAS Speed). The goal of the algorithm is to choose an action from the action space to either solve a conflict, or which leads to the latest start time of a conflict. Furthermore, they extend it to include heading changes as well (henceforth referred to as MAS Heading). Differently from their work, we consider the speed changes separately from heading changes, and do not combine the two different ways of regulating a MiT network. For a more detailed description of the algorithm, we refer the reader to [15].

Furthermore, we adapt the Modified Voltage Potential (MVP) conflict resolution algorithm proposed in [16]. The MVP algorithm models conflicting aircraft as identically charged particles that repel each other in such a way that the conflict does not occur. The result is a displacement vector that is used to compute changes in the speed of conflicting aircraft. We refer the reader to [16] for a detailed description. Finally, we also show the case where no intervention is made, for reference.

Figure 9 shows the MiT network we consider. In the case of MAS Speed, aircraft must decide between three actions: cruise, accelerate, decelerate. Such a decision must be made every 5 s and the algorithm is validated greedily for each aircraft in parallel. Acceleration and deceleration are fixed to $\pm 4000 \, \text{NM/m}^2$. For the MVP algorithm, whenever there is a conflict pair, speed changes are applied to solve the conflict.

The scenario of MAS Heading is shown in Figure 10. In this scenario, aircraft can be put on parallel track to enable them to cross the intersection conflict free. Each flow is divided into three tracks separated by 5 NM. Aircraft must choose which track they must take. Finally, the sub-flows are merged into a single outgoing flow.

In this section, we consider 4 different scenarios: 2 aircraft per flow, 4 aircraft per flow, 8 aircraft per flow and 16 aircraft per flow. Four intervention methods are examined: MAS Speed, MAS Heading, MVP and a reference case (no intervention). Figure 11 shows the indicators for the case with 2 aircraft per flow. When an action is chosen to reduce the effect of the conflict, the CC and NND scores drop dramatically. This result indicates that our metric might be able to capture perceived controller complexity. Nevertheless, we see a huge spike for edge density and strength in the case of MAS Speed. This is attributed to the fact that the algorithm is not able to solve this conflict. The MVP algorithm has a similar ED score to when there are no interventions, but the strength score behaves differently, which is due to the fact that the conflict is solved once it is detected. This is further confirmed from the comparison between MAS Speed and MVP.

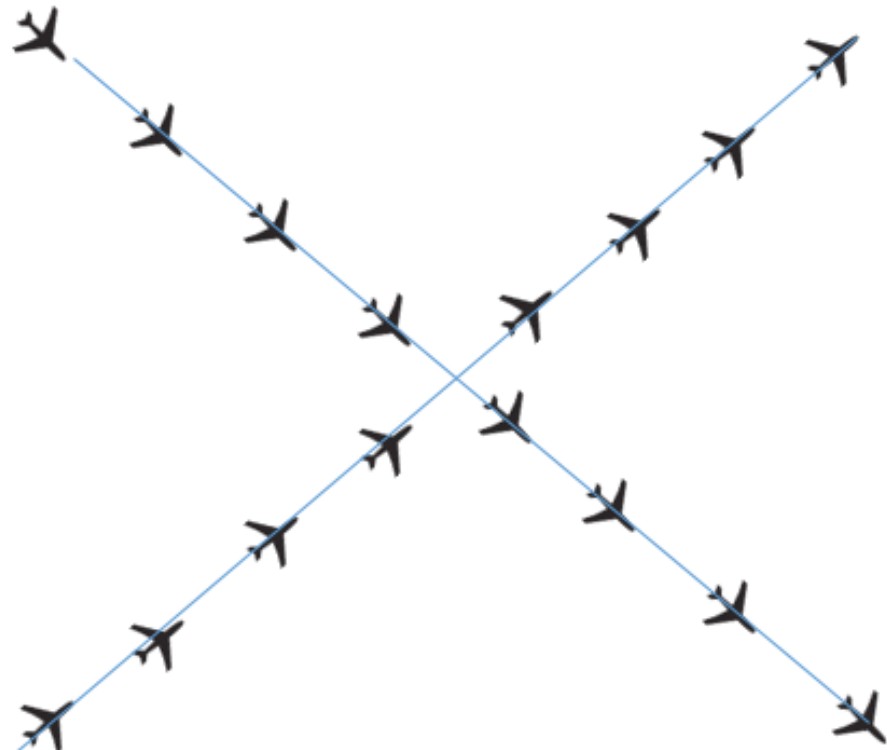

**Figure 9.** Miles-in-Trail scenario for the case of speed changes.

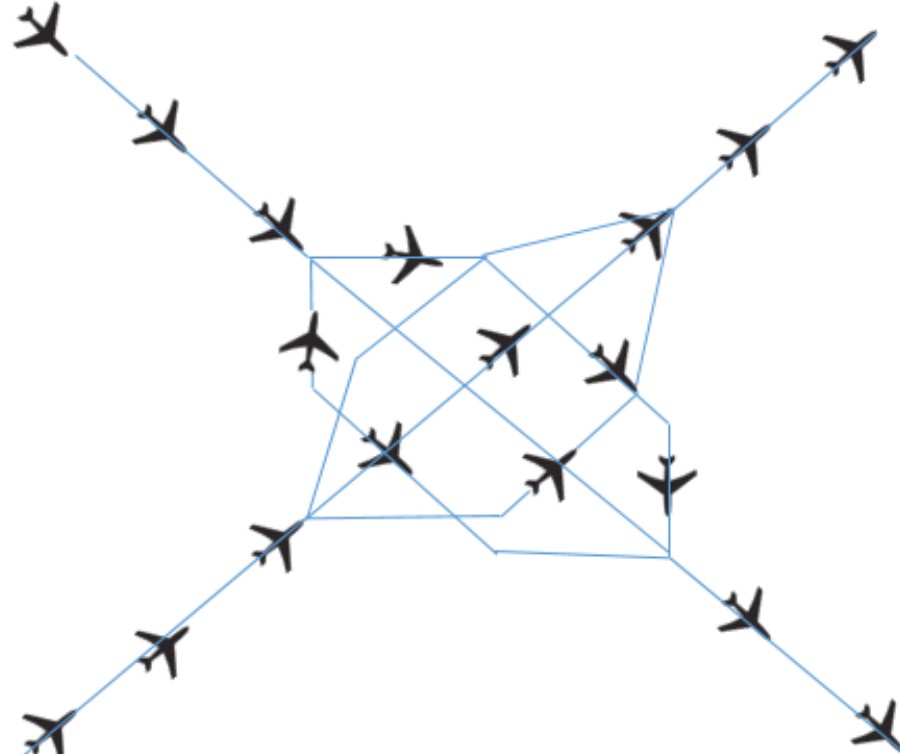

**Figure 10.** Miles-in-Trail with heading changes.

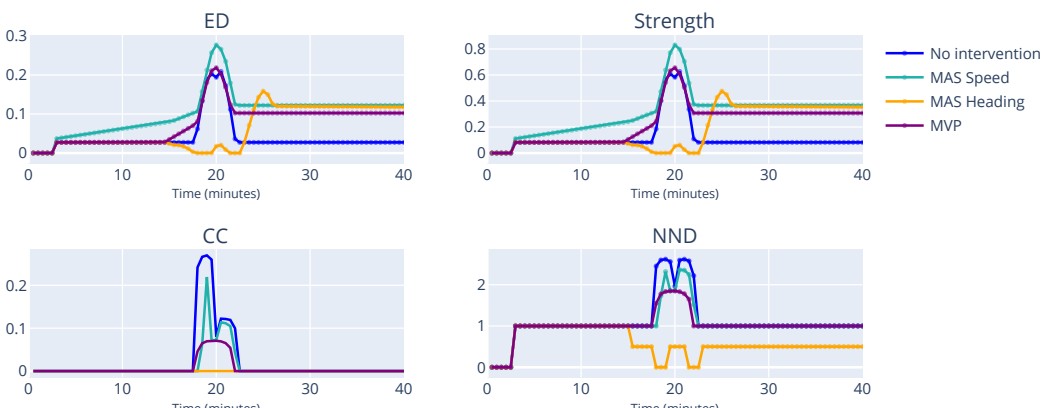

**Figure 11.** Miles-in-Trail with 2 aircraft per flow.

Figure 12 shows the indicators for the case of four aircraft per flow. The results here are similar to the previous case. However, we note that in this scenario, MAS Speed does not lead to a spike in ED and strength. Nevertheless, MVP leads to a higher ED and strength score. MAS Heading leads to a more complex solution of the MiT scenario when four aircraft per flow are present, with only the NND not resulting in a spike.

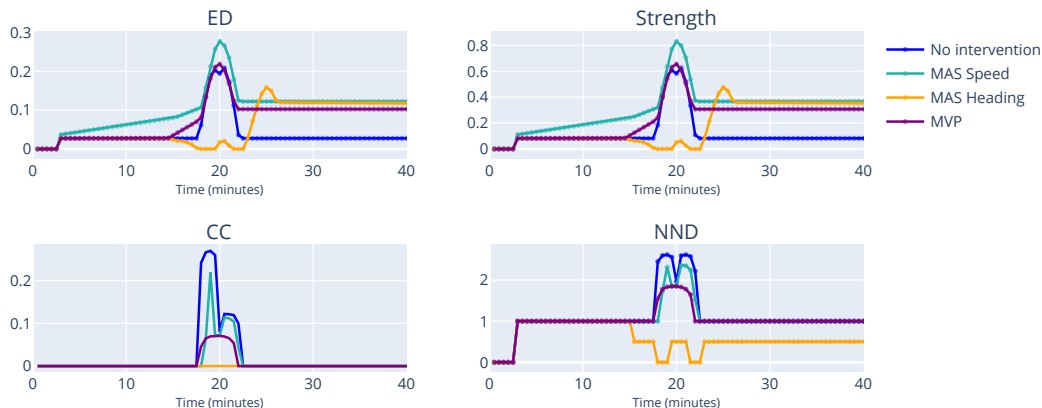

**Figure 12.** Miles-in-Trail with 4 aircraft per flow.

Figures 13 and 14 provide similar information. In these two scenarios, where the number of aircraft per flow is quite high, the speeding changes algorithms lead to a noticeably lower scores for CC and NND. With many present aircraft, changing the heading might lead to a more chaotic situation. There is a bigger possibility of aircraft creating interdependencies with other aircraft that they would not have if the traffic was regulated by speed changes. Furthermore, we note that in all scenarios, the spike in the MAS Heading algorithm happens later then in the case of MAS Speed and MVP. This is an indicator that the bigger interdependencies happen when traffic is rerouted to its original flow. The indicators provide elaborate information to controllers, as they can be alerted that a complex situation will start. In situations with many present aircraft, heading changes are more difficult for a controller to handle, which is further evidence that our indicators are suitable to be used in real traffic. This analysis shows that the indicators can be used to assess how conflict resolution algorithms affect the complexity of the airspace, which is a dimension traditional CR algorithms do not consider.

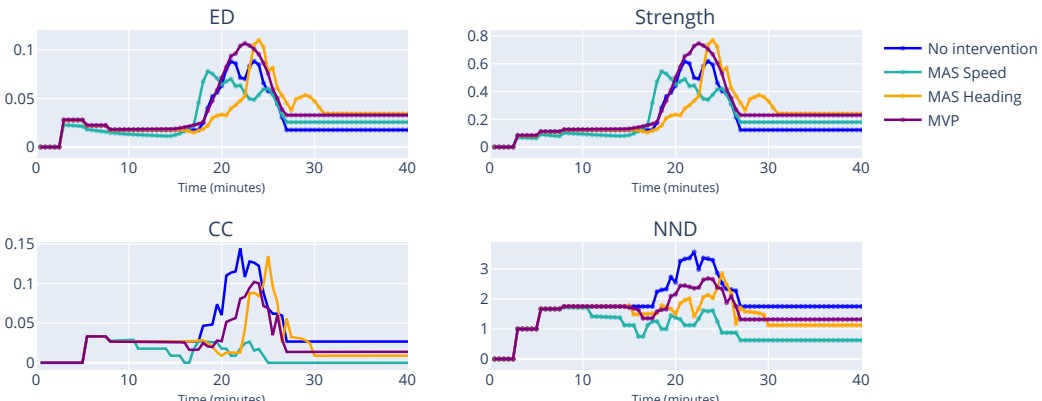

**Figure 13.** Miles-in-Trail with 8 aircraft per flow.

Moreover, the proposed indicators can be used to decide appropriate solution techniques for given scenarios. For example, in cases with fewer aircraft, heading changes lead to lower complexity, while in scenarios with many aircraft, speeding changes might be preferred. Finally, the indicators can be used to design novel conflict resolution algorithms. Such algorithms can be used to not only solve conflicts, but also improve the quality of solutions. For instance, an algorithm based on the indicators could propose solutions that at best improve the complexity of the sector and at worse do not increase complexity, in addition to solving the present conflicts.

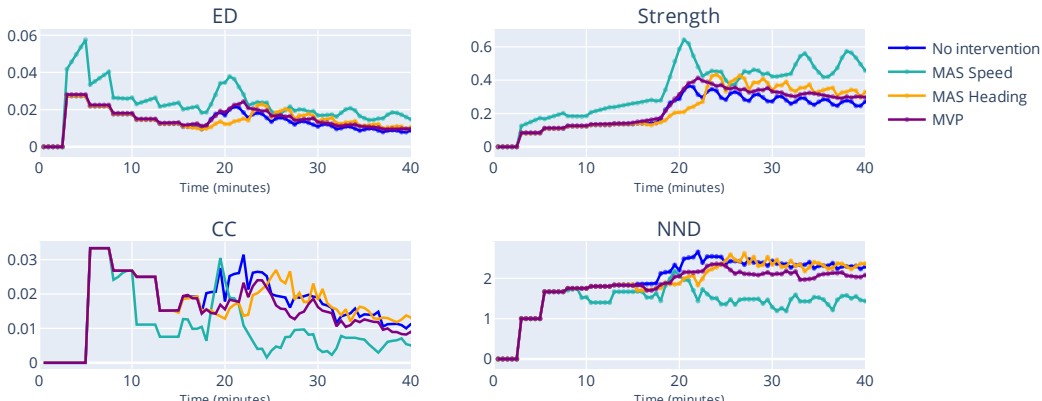

**Figure 14.** Miles-in-Trail with 16 aircraft per flow.

## 6. Evaluation on Real Traffic

### 6.1. Data

We evaluate these indicators on a day of flights taken from DDR2 (m3 file format consisting of actual trajectories). We take sector configurations of that day and eventually end up with 178 sectors, all in the European Civil Aviation Conference (ECAC) (https://www.ecac-ceac.org/, accessed on 23 November 2021) area. Sector borders are kept unchanged throughout the day. The flights and sectors that are used in this work are en-route, i.e., active at 25,000 feet and higher.

In order to determine the horizontal and vertical thresholds through which we define interdependencies, we do a system wide analysis of pairwise aircraft horizontal and vertical distances. As such, for each individual sector, we compute the average pairwise horizontal and distance based on data from the previous day. The thresholds are then set as the mean pairwise distance (horizontal and vertical). In this way we adapt the thresholds for each sector, expecting that historical data can be useful to identify some traffic features if the operational context is similar. To those distances, we add as a buffer a value of 5 NM horizontally and 1000 feet vertically. Interdependencies should not be defined universally,

as sector size plays a role in the distances or aircraft. Using such a method, we ensure that horizontal and vertical thresholds are set for each sector differently.

To get the complexity scores for the indicators, we sample the data every 30 s for the whole day.

### 6.2. Complexity Indicators

In this section, we visualize the complexity indicators for an hour of operations. The sector we chose had, on average, the most aircraft present at the same time. In Figure 15, we also visualize the occupancy in the the sector at each time step. This can be thought of as an extension of this metric into the time domain. Occupancy is one of the simplest yet most used classical complexity metrics. This section presents a comparison of this metric with the proposed indicators and we show that occupancy cannot capture nuances in aircraft interdependencies and thus complexity.

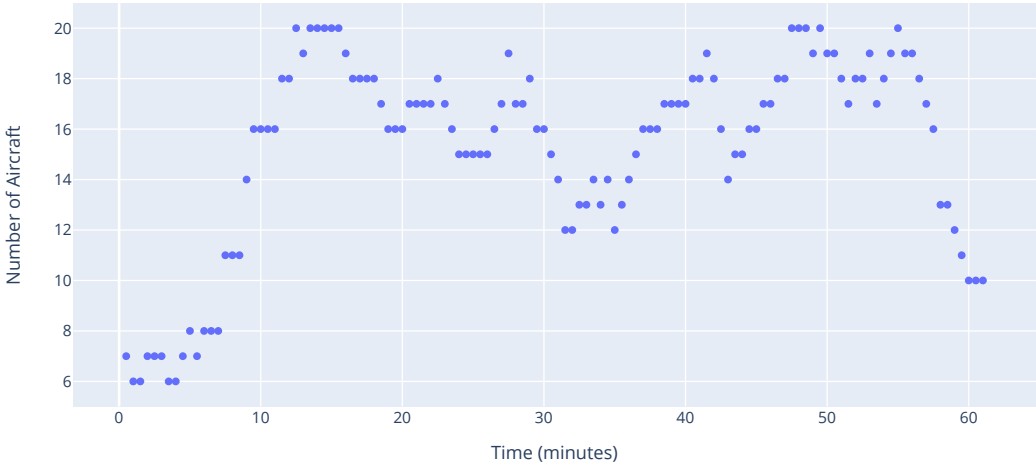

**Figure 15.** Number of flights present in the sector.

In Figure 16, ED of the hour of operations is shown. The ED score is relatively low, with the highest being around 0.6, which occurs when the number of aircraft is around 8. This means that the majority of aircraft do not get close enough to each other to be considered interdependent. Such a result further confirms initial statement, that the number of aircraft does not present a full picture of complexity.

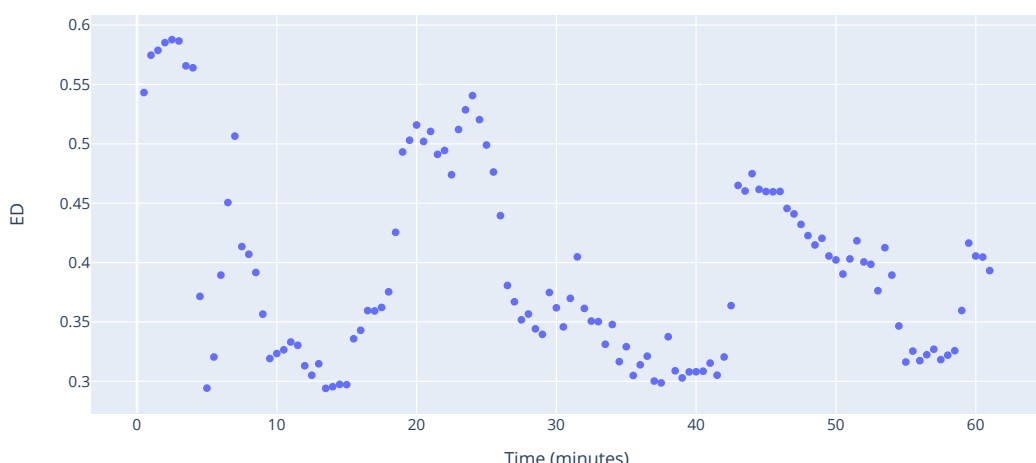

**Figure 16.** Edge density.

Figure 17 shows the clustering coefficient throughout the hour. The overall distribution is similar to that of ED, which suggests that there is an area in the sector where aircraft

tend to cluster. Nevertheless, one can see that such clusters do not last long (in this case around 5 min).

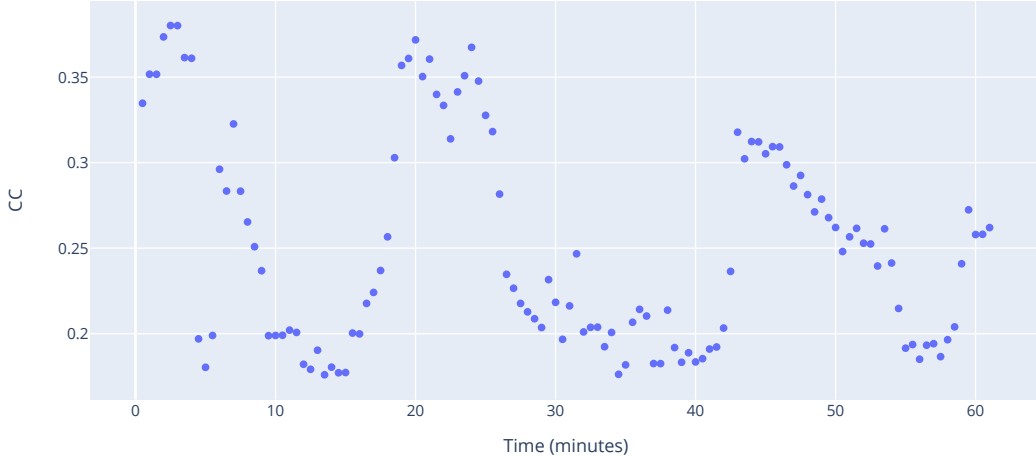

**Figure 17.** Clustering coefficient.

The NND score, shown in Figure 18, provides similar information. There is a peak around the 20 and 50 min marks, which corresponds to a peak in the other indicators as well as the number of aircraft. However, in the beginning, when the CC and ED score quite high, NND is at its lowest. This is due to a small graph size, which is further supported by a small number of aircraft. Between the two peaks, a steady decline and increase of NND is observed. Such a behaviour is evidence of the shape of the graph during that time, meaning that even though clusters are not formed, interdependencies are not just pairwise, but span multiple aircraft. This shows that relatively complex situations can arise even when the graph is small (as evidenced by ED), with a group of aircraft needing special attention.

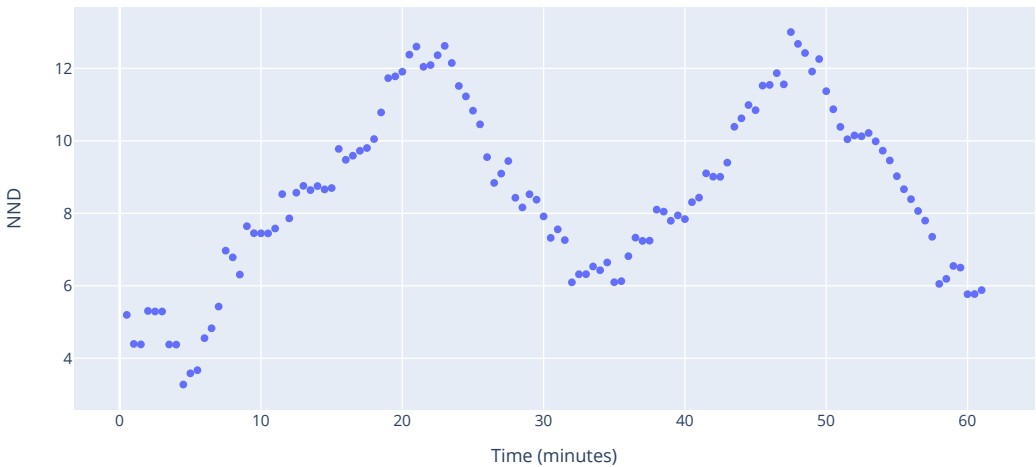

**Figure 18.** Nearest neighbour degree.

The strength indicator for the hour of operations is visualized in Figure 19. Similarly to the rest of the indicators, there are peaks around the 20 and 50 min marks. However, the shape of the distribution is akin to that of NND, which supports our claim in the previous paragraph about the nature of interdependencies. On top of that, the similarity between NND and strength further suggests that in this particular case, not only do interdependencies span multiple aircraft, they are also quite strong, which adds to the complexity of the situation.

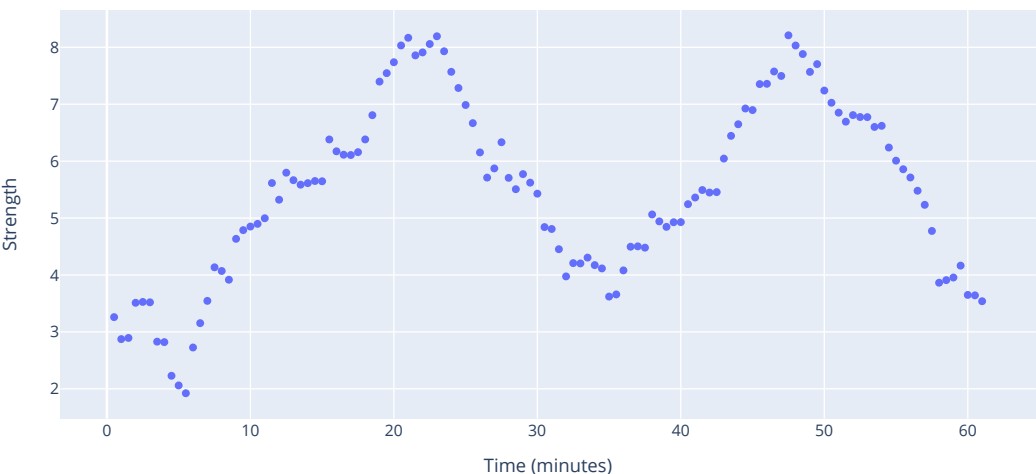

**Figure 19.** Strength.

*6.3. Correlation between Number of Aircraft and the Indicators*

In this section, we show that the combination of the proposed indicators can capture more detailed information than the number of aircraft, which is the simplest complexity indicator. To do this, we calculate the Pearson correlation coefficient for all sectors. Table 3 shows the mean correlation for each indicator.

**Table 3.** Mean correlation of each indicator with the number of aircraft.

| ED | Strength | CC | NND |
|:---:|:---:|:---:|:---:|
| 0.39 | 0.86 | 0.64 | 0.88 |

We notice that the Pearson correlation coefficient for ED is a value of 0.39 which means there is low correlation. This result is further evidence that number of aircraft does not account well for sector complexity.

The correlation of aircraft count and CC and NND is higher. Especially, in the case of NND the correlation coefficient is more than 0.88, indicating high correlation. This behaviour is consistent with results from the previous section, which means that when there are more aircraft in the sector, interdependencies tend to span multiple aircraft, while being less likely to form clusters.

A relevant result is that CC correlates to the number of aircraft more than ED. When a sector is more densely populated, clusters tend to form even when only a fraction of present aircraft are interdependent. This shows that even when a high number of aircraft correlates with more complex situations, our indicators give more information about the nature of this complexity.

As previously stated, the evolution of strength through time demonstrates that when there is a high number of aircraft present in the sector, interdependencies tend to be stronger. The results shown in Figure 19 verify this claim.

*6.4. Correlation between Indicators*

In order to show that all indicators are needed, we calculate how they correlate with each other and interpret these results, shown in Table 4.

We notice that correlation between CC and ED is low, an expected result from the results of the previous section. This is additional evidence that even small graphs can be quite complex. The mean correlation between ED and NND serves are further confirmation.

The correlation between NND and CC is high. Both of these indicators are defined to give higher scores to more interconnected graphs, thus this is an expected result. Nevertheless, these indicators inherently account for different topologies, as evidenced by Figures 17 and 18.

**Table 4.** Mean correlation between indicators.

|  | ED | Strength | CC | NND |
|---|---|---|---|---|
| ED |  | 0.65 | 0.54 | 0.61 |
| Strength | 0.65 |  | 0.83 | 0.98 |
| CC | 0.54 | 0.65 |  | 0.82 |
| NND | 0.61 | 0.98 | 0.82 |  |

The results indicate low correlation between the ED and strength indicators. As we have noted before, graph size (which ED measures), shows that not all aircraft present in the sector contribute to the overall complexity. Strength adds a layer of information by measuring the severity of interdependencies. The strength-ED correlation implies the presence of bigger graphs with loosely interdependent aircraft and smaller graphs with tightly interdependent aircraft.

The strength-NND and strength-CC correlations are quite high. This suggests that in the presence of tighter interdependencies, clusters and other forms of multi-cluster interdependencies are very likely to form. However, as shown in the MiT example, very different geometries with the same strength can emerge. Therefore, to capture such topological information and combine it with the severity of interdependencies, it is important to consider all indicators.

## 7. Conclusions and Future Work

In this paper, we formalize four indicators based on graph theory to measure sector complexity. These indicators combine topological information gathered from interdependency geometries with the severity of interdependencies to present a full and nuanced picture of complexity. Furthermore, we consider the evolution of complexity in time and do not simply give one single sector complexity score. Simulation results indicate that the proposed indicators give detailed information on complexity and overcome drawbacks of existing metrics.

We evaluate the indicators in synthetic traffic geometries, as well as in a standard situation such as Miles-in-Trail. For the latter, we adapt several conflict resolution algorithms and show how different resolution strategies can affect the overall complexity. We argue that complexity is a factor conflict resolution algorithms should consider and through results show different strategies might be preferred depending on the number of aircraft.

Furthermore, the proposed indicators are assessed using a day of flights taken from DDR2 data. Sector configuration of that day is taken and we monitor 178 sectors in the ECAC area. Results show that the indicators are more effective in capturing geometrical complexity than aircraft density. The information obtained from using the indicators can provide the controllers with a better understanding of complex areas inside the sector. On top of that, we demonstrate that the four indicators express different facets of complexity, confirming that all indicators are needed.

The indicators are evaluated using real traffic by taking a one hour window with samples every 30 s which results in a stable output. Nevertheless, the length of the time window must be investigated further. Several factors must be considered when deciding on the length of the window. One of the most important factors is including uncertainty in trajectory prediction, which we do not consider in this work. As such, suitable time windows might be smaller, or might require different sampling strategies.

As previously stated, the indicators provide a new framework in the design of conflict resolution algorithms in order to preserve safety while reducing traffic complexity. Complexity is usually not taken into account in designing such algorithms, which can lead to solutions that can make the situation difficult for the controller to handle. Additionally, by using the indicators, algorithms can be tuned to encourage resolutions that lead to lower complexity and discourage those that increase complexity. Such algorithms can be used to improve the quality of resolutions, in addition to solving the present conflicts. For instance, Isufaj et al. [35] propose a multi-agent reinforcement learning approach to conflict

resolution which considers airspace complexity as one of the factors that the model must optimize in addition to solving conflicts. The indicators proposed in this work could allow for a more granular optimization of complexity by providing more detailed information.

In addition, by modelling traffic as a graph, we open the door to further applications of graph theory in aviation. First of all, this work should be a baseline for designing complexity indicators based on graph theory. As such, future work should consider different methods for the definition of interdependencies and the value of thresholds. For instance, machine learning methods can be used to generate the graph [36]. Furthermore, U-Space [37,38] is envisioned to be a set of services to provide ATM to unmanned aviation in Europe. One of its key services will be dynamic capacity management [37], used to for strategic conflict resolution. The indicators proposed in this work, can be adapted to this use-case and serve as an implementation of this service.

Finally, a very important continuation would be the mapping between the indicators and controller workload, which is an ongoing topic in the ATM community. Ways of measuring the workload could be through subjective scores or more sophisticated methods such as EEG [12,13]. Such a work would investigate if the proposed indicators are a good predictor of the measured workload. Last but not least, it is very important to consider how the information provided by the indicators should be presented to the controllers. In this work, we make a simple attempt by showing the interdependencies and the indicator scores at various points in time. However, more research is needed which takes into account controller preferences and other various factors.

**Author Contributions:** Conceptualization R.I., T.K. and M.A.P.; software, R.I.; methodology R.I., T.K. and M.A.P.; investigation R.I., T.K. and M.A.P.; supervision M.A.P.; original draft preparation, R.I.; review and editing, R.I., T.K. and M.A.P. All authors have read and agreed to the published version of the manuscript.

**Funding:** The first author has received funding from the SESAR Joint Undertaking under the European Union's Horizon 2020 Research and Innovation Programme under grant agreement No 783287. The opinions expressed herein reflect the authors' view only. Under no circumstances shall the SESAR Joint Undertaking be responsible for any use that may be made of the information contained herein.

**Institutional Review Board Statement:** Not applicable.

**Informed Consent Statement:** Not applicable.

**Data Availability Statement:** The data presented Section 6 are taken from DDR2 (m3 file format) and contains flights for the date 19 February 2019. The data presented in 5.2 can be downloaded following this link ( https://uab-my.sharepoint.com/:u:/g/personal/1558645_uab_cat/EWlcJReOROdDshssAojtWoAB-8sxaTxsPEghoME6dEElHw?e=DcR7rR accessed on 22 November 2021).

**Conflicts of Interest:** The authors declare no conflict of interest.

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
