# Peer review of "Spatiotemporal Graph Indicators for Air Traffic Complexity Analysis"

_aerospace, doi:10.3390/aerospace8120364_

Round 1

Reviewer 1 Report

The paper is nicely written. However, this can be improved:

  1. Elaborate more on the methods especially apparatus, algorithm, etc.
  2. What is the reason for choosing 8 or 16 aircraft?

Reviewer 2 Report

This paper presents a graphics-based approach to measuring air traffic complexity. The relationships between aircraft in the airspace are used for determining the edges of the network. Four indicators are proposed, including edge density, strength, clustering coefficient, and nearest neighbor degree. The approach is tested with syntenic aircraft trajectories and real traffic data. This work seems very interesting and the approach is promising. Here I listed major concerns that the authors may want to take into account for their next submission.

1. While the network is formulated based on the distance between aircraft. I would like to see the arguments about the sensitivity analysis, what’s the effect of different distances on the complexity indicators? There were several works on the study of air traffic complexity based on network approach, for example, aggregated networks and temporal networks formulated from air traffic controller’s communication data. It might be interesting to compare these works with the presented work.

2. If possible, could you please provide a comparative analysis between the four complexity indicators with previous complexity indicators? or maybe with the mental workload or controller’s subjective rating? I know that the proposed complexity indicators are time-dependent. 

3. A geographical map showing the scope as well as the complexity would be great.

Reviewer 3 Report

This paper is a good one and is worth to be published. I have only minor comments that are detailled below:

p5, l280 : The definition of the weights make sense, however it relies on the choice of the two thresholds for horizontal and vertical distances. The tuning procedure for this parameter is not clear to me and can be made more explicit in the text. In particular, can we expect some kind of stabilization within a given interval of values ?

Section 4: All indicators are based on positions, not speed vectors. From a graph theoretic point of view, indicators based on Laplacian operator and its variations may have been considered. Do the authors plan to implement this kind of compexity metric in the future ?

Section 6: As usual with real traffic, a comparison with perceived worload is of interest, altough having access to expertized traffic samples is not so easy. Do the authors plan to perform such a comparative study ? 

A really minor point: p2, l38 "immense" -> "huge" ?
